

# The response of small boreal catchments to extreme weather event: Hurricane Larry

Kavi M. Heerah[1,2], Kailee Clarke[2], Heather E. Reader[2]

[1]Environmental Sciences Program, Memorial University of Newfoundland and Labrador, St. John's, A1C 5S7, Canada
[2] Department of Chemistry, Memorial University of Newfoundland and Labrador, St. John's, A1C 5S7, Canada

*Correspondence to*: Heather Reader (hreader@mun.ca)



**Abstract**

The boreal environment is high in dissolved organic carbon (DOC) and iron concentrations. This DOC is enriched in functional groups allowing it to bind strongly with iron and act as a significant source of iron to the coastal and marine environment. As climate change intensifies more extreme weather events will affect the northern hemisphere and boreal environment. These weather events can lead to massive fluxes of material from the environment but the impact it will have on the boreal environment is currently unknown. Hurricane Larry made landfall on Newfoundland (NL) in 2021 providing

an opportunity to investigate how the boreal environment will react to extreme weather events. We sampled three rivers before and after the hurricane to see how DOC and iron concentrations, lability, and colour were affected by the hurricane. We found that a high percentage of forest and peatlands buffered against increases in DOC and colour, with wetlands buffering an increase in iron concentrations. This study represents one of the first to observe boreal catchment responses to

extreme weather events such as hurricanes.

**Short Summary**

The boreal environment has been highlighted as an important source of carbon and iron to the ocean. Hurricanes rarely reach the boreal environment, but with climate change hurricanes can reach further north affecting carbon and iron export from this region. This paper sampled three rivers before and after a hurricane. We found the catchments buffered increases in carbon

and iron from the hurricane. This study represents one of the first to investigate hurricanes in the boreal environment.

**1 Introduction**

Dissolved organic matter (DOM) is a complex mixture of compounds of diverse biological origins and diagenetic states. Rivers deliver significant amounts of terrestrial DOM and other important macro- and micro- nutrients to the coastal ocean, globally (Charette et al., 2020; Hedges et al., 1997; Kritzberg et al., 2014). Terrestrial DOM impacts the coastal

ecosystem by acting as a substrate for microbial degradation processes, and limiting light penetration, tipping the balance towards net heterotrophy (Jennings et al., 2010; Jiao et al., 2011; Reader and Miller, 2014; Ward et al., 2017). Terrestrial DOM brings with it a number of important nutrients, and in particular is recognized as an important supply of dissolved iron (dFe), thereby increasing coastal productivity (Krachler et al., 2010; Kritzberg et al., 2014).

High intensity storm events such as hurricanes have been shown to cause increases in terrestrial DOM loading in

rivers and estuaries (Asmala et al., 2021; Austnes et al., 2009; Tang et al., 2019; Yan et al., 2020). A number of studies have found that significant amounts of a river's annual DOM load can be exported as a result of a major storm event, in some cases up to 50% or more (Austnes et al., 2009; Avery Jr. et al., 2004; Clark et al., 2007; Raymond and Saiers, 2010; Yang et al., 2015). These massive fluxes of DOM occur over short time periods, and can have drastic consequences for coastal





carbon cycling (Asmala et al., 2021; Bukaveckas et al., 2020; Hounshell et al., 2019; Yoon and Raymond, 2012). Yan et al (2020) found that the DOM flux associated with Hurricane Harvey in 2017 increased carbon cycling in Galveston Bay, Texas (USA) with the majority of the flux mineralized within a month of the storm. On the other hand, Avery et al (2004) found that the majority of the riverine DOM associated with Hurricane Floyd in 1999 (Long Bay, North Carolina, USA) increased the concentration of recalcitrant DOM in Long Bay. Regardless of the fraction of bioavailable DOM for each event, the total fluxes of DOM and freshwater are such that the coastal bacterial community is altered (Bukaveckas et al., 2020; Osburn et al., 2019; Yan et al., 2020). The impact of these storm events persist in the coastal environment for long periods of time from days to months (Asmala et al., 2021; Avery Jr. et al., 2004; Yan et al., 2020).

Catchments in Newfoundland, Canada are dominated by peat barrens and boreal forest (Latifovic et al., 2017), both rich in organic carbon and iron. DOM from boreal and peatland ecosystems have been identified as important for biogeochemical nutrient cycling (Krachler et al., 2010; Kritzberg et al., 2014; Worrall et al., 2006). This DOM can form complexes with iron more resistant to flocculation allowing greater terrestrial inputs into the coastal environment (Heerah and Reader, 2022; Herzog et al., 2020; Krachler et al., 2015; Kritzberg et al., 2014). DOM concentrations have been increasing in inland waters across the boreal region due at least in part to warming conditions and increased precipitation, leading to brownification of waterways and increasing terrestrial inputs to the coastal ocean (Björnerås et al., 2017; Evans et al., 2005; Kritzberg and Ekström, 2012; Lepistö et al., 2021; Roulet and Moore, 2006; de Wit et al., 2016). With climate change and associated warming of the surface ocean, the intensity and frequency of storms in the North Atlantic are predicted to increase, with hurricanes reaching farther north and potentially affecting boreal regions more regularly. (Bender et al., 2010; Holland and Bruyère, 2013; Mann and Emanuel, 2006; Paerl et al., 2019; Smith et al., 2010). The impact of the convergence of these two phenomena is currently unknown, but clearly has the potential to impact the coastal waters of the North Atlantic.

In September 2021, Hurricane Larry made landfall on the Avalon Peninsula of Newfoundland, Canada as seen in Fig 1. This is the first hurricane within 11 years to reach the shores of Newfoundland, the previous one being Hurricane Igor in 2010.



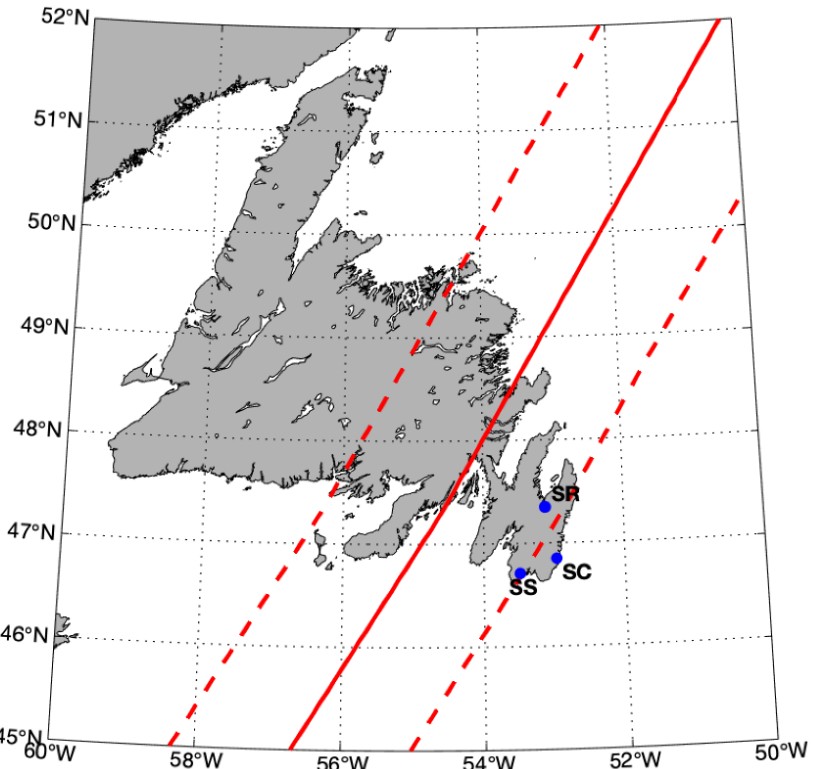

**Figure 1: Path of Hurricane Larry (solid line) with extent of maximum winds (dashed lines) and sampling locations (blue dots).**
**Hurricane path and wind extent data from the National Hurricane Center Atlantic HURDAT2 database(Landsea and Franklin, 2013). Map created using M_Map package for MATLAB (Pawlowicz, 2020) using the NOAA GSHHG high-resolution coastline database (Wessel and Smith, 1996)**.

While the island of Newfoundland experiences storm events with regular frequency, typically most of these events occur in the winter months, when productivity on land is limited and dissolved organic carbon (DOC) and iron concentrations are
relatively low (Bowering et al., 2023; Broder et al., 2016; Iavorivska et al., 2016; Rosset et al., 2020; Singh et al., 2014). With the predicted increases in hurricane activity in the Atlantic Ocean, the island can expect to experience more of these events during the warm summer months, potentially leading to considerably increased carbon and iron export to the coastal zone that would not be experienced in a typical winter storm event.

Taking advantage of the well-predicted path of Hurricane Larry which made landfall in close proximity to Memorial
University, we collected DOM samples from three rivers before and after landfall to better understand how the quality and quantity of DOM export is impacted by large scale storm events in this region. This is the first study of hurricane impacts on the carbon cycling in Newfoundland to the author's knowledge.



## 2 Methods

### 2.1 Sample Collection

Samples were collected from three river sites on the Avalon Peninsula of Newfoundland, Canada before (September 10th, 2021) and after the landfall of Hurricane Larry (September 11th, 2021). Hurricane Larry made landfall shortly after midnight local time (23:00 Newfoundland Standard Time, NST) on September 11th, 2021. Samples were collected 12-18 hours after landfall as soon as it was determined that the road was passable to all three sites. The three sites are equipped with water level and discharge gauges by ECC: Seal Cove Brook near Cappahayden, NL (SC, ECCC 02ZM009), St Shotts

River near Trepassey, NL (SS, ECCC 02ZN002), and South River near Holyrood, NL (SR, ECCC 02ZM016). Water level and discharge data was extracted from the ECCC Real-time Hydrometric Data Website(Real-Time Hydrometric Data - Water Level and Flow - Environment Canada, 2024) . Temperature and pH shown in Table 1 were measured in situ using a Thermo Scientific Orion Star A329 meter and pH probe.

### 2.2 Catchment Characteristics

The three catchments are dominated by boreal forest and peatlands, The three catchments occupy 2 distinct ecoregions, SR and SC are situated in the Maritime Barrens ecoregion and SS is situated in the Eastern Hyper-Oceanic Barrens ecoregion. The two ecoregions are distinguished by their proximity to the southernmost coast of the Avalon peninsula, while both regions are characterized by peat barrens and boreal tree species, the Eastern Hyper-Oceanic Barrens are exposed to strong winds, sea spray, and heavy fog most of the year. Tree vegetation in the region is limited to sparse patches of tuckamore (or

Krumholtz formations), which are low-lying shrub/thicket formations, as opposed to typical though somewhat stunted boreal forest found in the Maritime Barrens.

SC is the largest catchment at 53.6 km$^2$ with SR and SS following at 17.3 km$^2$ and 15.5 km$^2$, respectively. SR's catchment is 39.90% forest, 44.84% peat and 1.28% wetland. SC has the highest proportion of forest at 50.76% and has 33.63% peat and 1.95% wetlands. SS is dominated by peat at 62.96% and has 13.02% forest and 10.04% wetland. These three landcover types

are strongly associated with DOM and iron quality and quantity, and thus were chosen for statistical analysis of the changes in DOM before and after the Hurricane Larry (Clark et al., 2008; Hedges et al., 1997; Krachler et al., 2010; Kritzberg et al., 2014; Olivie-Lauquet et al., 2000; Weyer et al., 2018). Full landcover details can be found in the Government of Canada 2015 Land Cover of Canada report (Latifovic et al., 2017). River discharge for the year for each catchment was used to calculate a relative Richard-Baker flashiness index (RB index) (Baker et al., 2004). The RB index is a dimensionless

indicator of how quickly a river returns to baseflow and can be used to compare catchment responses to storm events.





### 2.3 Dissolved Organic Carbon

DOC was quantified as non-purgeable organic carbon on a Shimadzu TOC-L-CPH analyzer. The instrument was calibrated with a recrystalized acetanilide primary standard $C_8H_9NO$, Across Organics, 99+% pure).

### 2.4 Optical Properties

DOM absorbance was measured on a Cary 300 spectrophotometer (Agilent), from 200 – 800 nm, with 1 nm resolution in a 5 cm quartz cuvette. Absorbance was converted to decadal absorption coefficient by $a$ = A/0.05, where 0.05 is the pathlength of the cuvette in m. The absorption coefficient at 350 nm ($a_{350}$) was used as a proxy for chromophoric DOM (CDOM).

### 2.5 Iron quantification

Iron was quantified using the ferrozine method as outlined by Violler et al. (2000) and Kritzberg et al. (2014). Briefly, 2.5 mL of sample is mixed with 515 µL of hydroxylamine hydrochloride solution and 250 µL ferrozine reagent and is then heated to just below boiling for 10 minutes. This step reduces all Fe to Fe (II) allowing complexation to the ferrozine reagent. Once the sample is removed from heat it is allowed to cool for 90 seconds before the pH is adjusted using 200µl of basic ammonium acetate solution, and the resulting ferrozine-iron complex is measured by its absorbance at 562 nm. The 120 method is calibrated using $Fe(III)Cl_2$. The $Fe(III)Cl_2$ calibration ranges from 0 to 5µmols/L using 6 points. The 6 points are 0,0.25µmols/L, 0.5µmols/L, 1µmol/L,2µmols/L and finally 5µmols/L. This method allows for a limit of detection of 0.0841µmols/L.

### 2.5 Biological Oxygen Demand

Biological oxygen demand (BOD) was quantified as the difference in dissolved oxygen over 14 days incubation 125 (room temperature, dark). Dissolved oxygen was measured in triplicate 60 mL BOD bottles by Winkler titration. Samples were fixed and stored in the dark at room temperature prior to measurement, and all samples were titrated on the same day.

## 3 Results

### 3.1 Change in river state

The hydrographs as seen in Fig. 2, show that all rivers were in a period of stable baseflow for the 24 hours prior to the 130 hurricane, experienced a peak in discharge within hours of landfall, and then remained in a high flow state for the 24 hours after landfall. In all three catchments, water level and discharge increased following the hurricnae with discharge increasing by 235% in SR, followed by 70% for SC, and 40% for SS between the first sampling and the initial peak in discharge. In SC,





the discharge stabilized after the initial discharge peak which occurred ~5.5 hours after landfall of the storm. In both SR and SS, discharge continued to increase after the initial peaks (2 and 5 hours after landfall, respectively) for the 24-hour period

after landfall. While we were not able to sample at the time of the initial peaks, all rivers were sampled at considerably higher discharges after the hurricane than before.

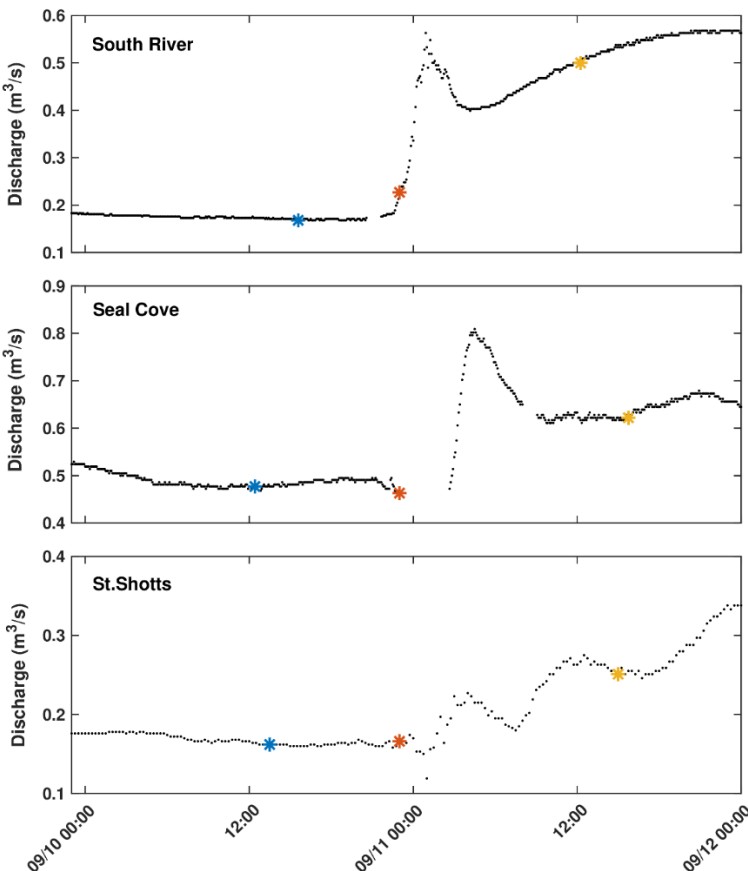

**Figure 2: Hydrographs of the three catchments for the 24 hours preceding and following landfall of Hurricane Larry. The blue and yellow asterisks represent the time (NST) of sampling prior to and after landfall, respectively. The**
**orange asterisk marks the time (NST) of landfall.**

Both temperature and pH (Table 2) exhibited statistically significant decreases in all three catchments between sampling times (t-tests, p = 0.0017, p=0.0499, respectively).

**Table 1: Physical characteristics of the three rivers before and after Hurricane Larry. Exact times of sampling**
**relative to landfall can be seen in Fig 2.**



| River | Time | Discharge (m³/s) | Temp (ºC) | pH |
|---|---|---|---|---|
| **South River (SR)** | Before | 0.168 | 20.5 | 7.16 |
| | After | 0.500 | 19.4 | 6.82 |
| **Seal Cove (SC)** | Before | 0.477 | 20.3 | 7.27 |
| | After | 0.622 | 19.7 | 7.06 |
| **St Shotts (SS)** | Before | 0.162 | 20.3 | 7.18 |
| | After | 0.251 | 19.5 | 6.61 |

Increases in DOC and iron concentrations were seen in all rivers, as well as increases in $a_{350}$ indicating increasing water colour as seen in Fig.3. None of these differences were statistically significant over all three rivers likely due to the small sample size and initial differences in conditions between rivers. However, increases in DOC concentrations for SR and SS were substantial (+237 µmol/L and +159 µmol/L, respectively) with corresponding increases in $a_{350}$ (+4.55 m$^{-1}$ and +3.44 m$^{1}$, respectively). Increase in iron was highest in SR (+1.57 µmol/L), and nearly the same in SC and SS (+1.01 µmol/L and +0.96 µmol/L, respectively).





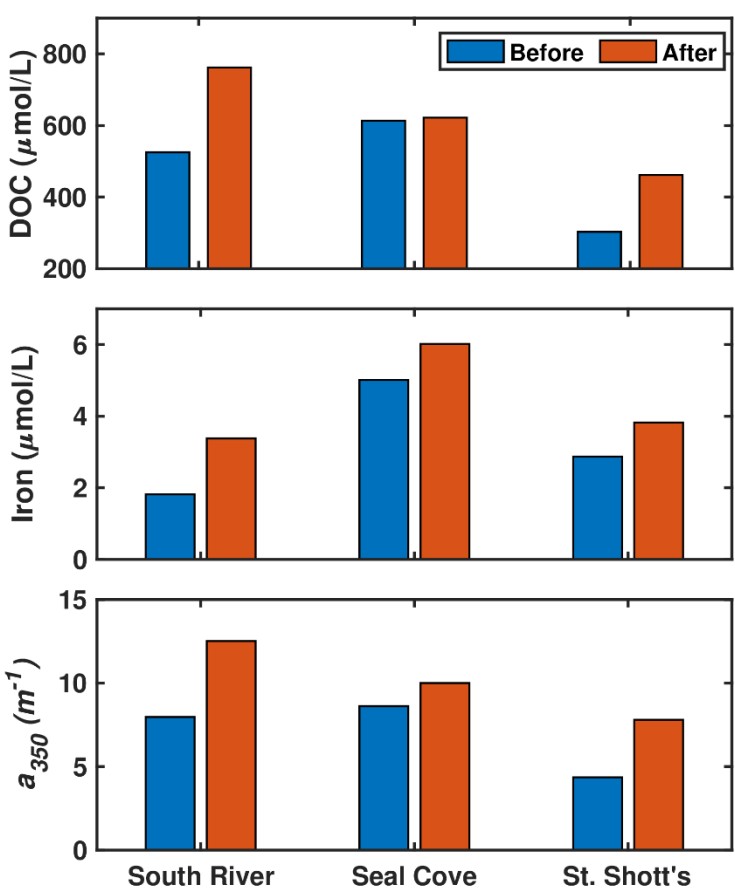

**Figure 3: DOM properties and iron before and after Hurricane Larry. Top, dissolved organic carbon (µmol/L); middle, iron (µmol/L); bottom, a350 (m⁻¹).**

## 3.2 Catchment characteristics and its effect on river conditions

Correlation analysis (Table 2) on the changes pre- and post- hurricane revealed that changes in DOC and $a_{350}$ (i.e. ΔDOC and Δ$a_{350}$) both had relatively strong negative correlations with the percent of peat in the catchment, as well as the total catchment area, and slightly less strong negative correlations to the percent forest cover. None of the correlations in the DOM analyses were statistically significant, due to the small dataset and low degrees of freedom, however the ΔDOC vs peat cover and $a_{350}$ vs peat cover approached statistical significance (p = 0.111 and p = 0.117, respectively). Neither ΔDOC nor Δ$a_{350}$ correlated with the amount of wetland in the catchment, with smaller negative relationships and much higher p values (0.686 and 0.681, respectively).



In contrast, changes in iron (ΔFe) showed the opposite behaviour, with no relationship to peat, forest or total catchment area, but a stronger negative relationship to the amount of wetland in the catchment, again starting to approach statistical significance with p = 0.181. Change in discharge was negatively correlated to amount of wetland in the catchment (r=-0.998, p=0.097), and had no relationship to any other landcover variables (p>0.800). High variability between BOD replicates limited the ability to assess bioavailability of the DOM accurately and precluded assessment of any changes before and after the hurricane. However, over all BOD samples appeared to be no more than a few percent of the total DOM.

**Table 2: Pearson's correlations between changes in DOM properties and catchment land cover types.**

|  | Peat | Forest | Wetland | Catchment Area | RB index |
|---|---|---|---|---|---|
|  | $\rho$ (p-value) | $\rho$ (p-value) | $\rho$ (p-value) | $\rho$ (p-value) | $\rho$ (p-value) |
| **Iron** | -0.567 | -0.258 | -0.960 | -0.385 | -0.569 |
|  | (0.616) | (0.834) | (0.181) | (0.748) | (0.614) |
| **DOC** | -0.985 | -0.869 | -0.473 | -0.928 | 0.187 |
|  | (0.111) | (0.329) | (0.686) | (0.243) | (0.881) |
| $a_{350}$ | -0.983 | -0.865 | -0.481 | -0.925 | 0.178 |
|  | (0.117) | (0.335) | (0.681) | (0.249) | (0.886) |
| **BOD/DOC** | -0.866 | -0.648 | -0.745 | -0.745 | -0.160 |
|  | (0.333) | (0.551) | (0.465) | (0.465) | (0.898) |

## 4 Discussion

Hurricane Larry made landfall in the late summer/early autumn of Newfoundland, a period where storms are not expected and where the productivity on land is peaking (Bowering et al., 2023). Following Hurricane Larry, all catchments experienced an increase in discharge, water level, and carbon and iron concentrations which is in line with current literature (Abesser et al., 2006; Austnes et al., 2009; Clark et al., 2007; Hounshell et al., 2019; Vaughan et al., 2019) but the magnitude of change as well as the time it took to return to baselevel varied. The difference in flashiness is expected based on the size, land cover differences and catchment complexity (Baker et al., 2004). SC, as the largest of the three catchments was the least





flashy incorporating the massive increase in precipitation.SR had the highest increases in fluxes of material following landfall highlighting that in addition to catchment size, landcover plays a major role in catchment response.

Export of material following hurricane Larry is buffered by the natural land cover present in all three catchments
(Fasching et al., 2019; Worrall et al., 2004). The peat and forest appear to be the dominant source of DOM and colour to the waterways with iron being supplied by wetlands in the catchments. SS has more peat and wetland proportionally than the two other catchments. As the type of land cover increases there is a reduced $\Delta$ for the associated property. $\Delta$TOC is buffered by an increase in the amount of peat present, a similar trend being seen for $\Delta$a350, and $\Delta$BOD/TOC. The buffering effect may be due to having a steady input of DOC from the peat layers. This buffering has been explored in past studies (Broder et
al., 2017; Broder and Biester, 2015). As these layers are capable of absorbing large quantities of water and are known to produce high levels of carbon, they are able to facilitate the extra water without significantly changing their export of DOC (Broder et al., 2017). Typically, the deeper layers of peat provide DOM to the waterways through groundwater infiltration with the top layers only being accessed during storm events and overland flow (Broder et al., 2017; Clark et al., 2007). With Newfoundland's shallow soil and high-water table levels (P.K Heringa, 1981; Price, 1992), the baseflow is expected to be
more dominant . Excess water will only act to speed up the export from the topsoil rather than access normally disconnected pools as in other catchments (Austnes et al., 2009). In systems such as these it is also common for a delayed peak to be observed where the water is absorbed and released later (Vaughan et al., 2019; Wagner et al., 2019)

Iron is strongly correlated with the proportion of wetlands present. Iron is found to build in such environments due to the saturated soils (Abesser et al., 2006; Neal et al., 2008; Olivie-Lauquet et al., 2000; Weyer et al., 2018). The iron which
can be built up in the reduced conditions found can be quickly transported as the storm increases connectivity (Abesser et al., 2006; Olivie-Lauquet et al., 2000). Similar to carbon, it is possible that with high water tables rather than access disconnected pools, the iron is more quickly mobilised leading to a higher concentration. In a catchment such as SS where the landscape is typically saturated a smaller increase may be seen as opposed to South River which comprises more forest areas.

Despite the increases being buffered by the presence of peat and wetlands the high concentrations present in these environments should not be ignored. In systems such as these it is also common for a delayed peak to be observed where the water is absorbed and released later (Vaughan et al., 2017; Wagner et al., 2019). The decrease of BOD/DOC is unexpected but can similarly be explained, with the more labile carbon being found in the upper layers of the soil(Fasching et al., 2019). As the topsoil is shallow, with a limited amount of carbon available for transport as the deep peat layer supplies more
carbon, the labile carbon does not change in quantity giving a smaller overall ratio. This can lead to the result where more peat, i.e., more baseflow, leads to less of an increase in properties seen.

The labile portion of the DOM is relatively small so the massive flux of carbon and iron occurring in these regions are most likely contributing to carbon storage on the coast of Newfoundland. Other studies have shown that the storm flux of DOM can either be mineralized or contribute to carbon storage based on the lability of the DOM and the microbial community
present (Asmala et al., 2021; Avery Jr. et al., 2004; Osburn et al., 2019; Yan et al., 2020). However, these studies are based





in bays and large estuaries where different biogeochemical processes can occur with a longer residence time. While small northern catchments such as these export high levels of DOM and Fe dismissing them can lead to underestimations in biogeochemical models (Khoo et al., 2023; Sanders et al., 2015). The direct export of material from small catchments into the coast is understudied.

The lack of continuous monitoring during the hurricane and the inability to capture the initial peak discharge following landfall may have prevented us from capturing the exact dynamics present (Rosset et al., 2020; Vaughan et al., 2017; Yoon and Raymond, 2012). An increase in labile protein like DOM has been observed in storm events being mobilized from the upper layers of peat (Austnes et al., 2009; Broder et al., 2016). This flush of labile protein can be seen in the first peak of outflow as it is quickly shunted to the coast. Due to sampling constraints, we were unable to capture the

initial peak, missing key information following a storm event. The DOM exported is also expected to change on the falling limb of the hydrograph (Vaughan et al., 2019).Without continuous monitoring of these sites during storm events small but significant changes in the dynamics of carbon and iron export may be missed.

## 5 Conclusions

We sampled three catchments on the Avalon Peninsula of Newfoundland before and after Hurricane Larry made landfall. All

three catchments are characterized by high levels of carbon and iron and all three increased their export following the hurricane. Our current results show that high intensity events such as Hurricane Larry will increase the export of material, but effects are buffered by saturated landcover. As climate change intensifies, it is likely that more hurricanes will affect the peat and boreal ecosystems found in the upper latitudes. Understanding how these environments will be affected by these intense storms will be pertinent as we consider the effects of global change on these ecosystems. The direct input of material

into the coast should be studied further with higher-resolution data collection as the frequency of storm events reaching Newfoundland increases. This report represents the first study of high northern environments reacting to hurricane.

## Data Availability

The data used in this manuscript can be accessed from Memorial University of Newfoundland Borealis database. The dataset

is currently available upon request but will be made open access upon final publication of the manuscript.



**Author contribution**

KH and HR were involved in the collection of samples, the design of the experiment, and the idea of the study. KH and KC carried out the experimental work. KH wrote the initial manuscript with major revisions conducted by HR and KC. All authors were involved in the interpretation of the data and editing of the manuscript.

**Competing Interest**

Authors have no competing interests to declare.

**Financial Information**

This study was funded by the Natural Sciences and Engineering Council of Canada (NSERC) Discovery Grant Program [RGPIN-2019–04947], and the Canada Research Chairs program.

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
