# Peer review of "The response of small boreal catchments to extreme weather event: Hurricane Larry"

_EGUsphere, 2024_

## Author Comment (AC1)

General comments: This manuscript was an interesting look at extreme weather (Hurricane Larry) on DOM in three small catchments in Atlantic Canada in 2021, a region which had not experienced such an event in about 10 years. The novelty here is that studies of hurricane effects on coastal ecosystems in this region of North America are rare but are valuable because of increasing frequency of intense Atlantic tropical cyclones that may reach the region. The coupling of DOM and Fe in this study was interesting. The streams were small such that even after passage of the hurricane, discharge <1 m3/s. Site information, methods all were good.

Specific comments: I found the results in Figure 3 and bore additional discussion. Though not significant in magnitude I wondered if the proportional increase in a350 was closer to the proportional increase in DOC or in Fe? This could indicate if the a350 is attributable to DOC or Fe or both.

*Reply: We have the Pearson's correlations between all the variables from our initial data analysis. For the purpose of the manuscript, we chose to only include the landcover correlations. The colour is mainly driven by the DOC seen by a correlation analysis between the $\Delta a_{350}$ and $\Delta DOC$ ($\rho=1.00$) compared to $\Delta a_{350}$ and $\Delta Fe$ ($\rho=0.708$).*

L160 - approaching significance isn't useful to describe; remove. In fact, the use of p values in this manner is not insightful and at times confusing. For example, Fe related to catchment properties - Why is $p = 0.181$ a relationship but $p = 0.616$ not a relationship? The relationship is evaluated by the strength of the correlation. While Fe had a high correlation with Wetland there was a lower correlation - not no relationship - to Peat.

*Reply: When discussing the significance of the relationship we were evaluating the $\rho$ values to assign importance. However, based on your comment I see how this is inconsistent, we have removed the discussion of p values in the results and amended the section to focus on the correlation strength with the caveat of high p values. Additionally, the p values were removed from Table 2. See new paragraph below spanning lines 155- 167(page 9)*

*"Correlation analysis (Table 2) on the changes pre- and post- hurricane revealed that changes in DOC and $a_{350}$ (i.e. $\Delta DOC$ and $\Delta a_{350}$) both had relatively strong negative correlations with the percent of peat in the catchment, as well as the total catchment area, and slightly less strong negative correlations to the percent forest cover. None of the correlations in the DOM analyses were statistically significant, due to the small dataset and low degrees of freedom. Neither $\Delta DOC$ nor $\Delta a_{350}$ correlated strongly with the amount of wetland in the catchment, with small negative relationships. In contrast, changes in iron ($\Delta Fe$) showed the opposite behaviour, with no relationship to forest or total catchment area, but a stronger negative relationship to the amount of wetland in the catchment and a weak negative correlation to peat. Change in discharge was negatively correlated to amount of wetland in the catchment ($\rho =-0.998$) and had no relationship to any other landcover variables. High variability between BOD replicates limited the ability to assess bioavailability of the DOM accurately and precluded assessment of any changes before and after the hurricane. However, over all BOD samples appeared to be no more than a few percent of the total DOM."*

L178 - Explain the last sentence in more detail.

*Reply: The last sentence refers to the fact that if catchment size was the only relevant factor for flux increase then SS as the smallest catchment should have the greatest increase, instead we see SR having this honour. The sentence has been amended to further explain this point (line 173-176, page 10):*
*"SC, as the largest of the three catchments was the least flashy incorporating the massive increase in precipitation. SR had the highest increases in fluxes of material following landfall despite being the second largest catchment. The dramatic increase in flux highlights that in addition to catchment size, landcover plays a major role in catchment response."*

L183 - What does "type of land cover increases" mean?

*Reply: The type of landcover here refers to the landcover type that is forest and peat for DOC and $a_{350}$ and wetlands for dFe. So, the sentence says as there is more coverage of a specific landcover type there is a reduced $\Delta$ for the associated property.*

*We have rephrased the beginning of the sentence (line 180, page 10) to improve readability: " as the percentage of landcover type increases…"*

L184 - Connect this point about buffering to the present work.

*Reply: Line 185 and 186(page 11) have been amended to further comments on the ability of the landscapes dominated by peat to buffer increases. :*
*"The landscape of NL is dominated by peat, and this can explain the small $\Delta$s seen in response to extreme weather events such as Hurricane Larry. Typically, the deeper layers of peat provide DOM to the waterways through groundwater infiltration with the top layers only being accessed during storm events and overland flow (Broder et al., 2017; Clark et al., 2007)."*

L193 - "found to build in" - rephrase to clarify

*Reply: Rephrased to "found to accumulate" to improve readability. (line 193, page 11)*

L196 - what might happen to the Fe once it encounters oxygen in surface waters and does this have any implication for DOC or CDOM?

*Reply: Once the Fe encounters oxygenated water a couple of different reactions might occur. The most dominant one would be oxidation from Fe (II) a soluble form to Fe (III) an insoluble form. This has a greater chance of flocculating and settling out. Some of the Fe (II) before being oxidized can form a complex with the DOM and thus remain in solution. While a further option is the Fe (III) can either form a complex with DOM or adsorb to DOM particles and thus remain in solution. In this case the adsorbed Fe and DOM can be disrupted by factors such as salinity and other ionic interactions. Fe can interfere with CDOM. One of the main effects Fe can have is increasing the absorbance of the CDOM and red shifting the fluorescence.*

L201 - who is ignoring them? Rephrase to clarify.

*Reply: Acknowledged see the amended sentence below (Line 200-201, page 11) :*
*"Despite the increases being buffered by the presence of peat and wetlands the high concentrations of DOM and Fe present in these environments requires ongoing research to understand the controls on their fluxes."*

L212 - Perhaps, it depends on the model. The systems here while having large DOC concentrations have small flows that offset their importance regionally. For South River, using average values reported here to upscale, I estimate the annual export of DOC is about 75,000 g C/yr. This is very small and probably not of regional importance but obviously could be important for local biogeochemical models. So, I would clarify that point.

*Reply: I believe the argument the sentence makes is still valid on the larger biogeochemical models. The argument here is that these small catchments are being neglected. While it is true this can have a larger effect on coastal and regional models, I believe the point still stands for large scale models as well. It is not the individual impact of the three rivers that make them worthwhile for consideration of larger biogeochemical models, rather the cumulative impact of many small rivers dotted across the boreal region each inputting 75,000 g C/ yr or more can add up. This can thus lead to underestimations preventing a full picture of global cycles. As Khoo et al. (2023) paper argues, small systems are the majority in these C-rich boreal environments, forming close to 90% of the systems by neglecting them in models it can cause an underestimation. Through a better understanding of carbon cycling, the global models can be improved upon which has been noted in Friedlingstein et al. (2014). Similarly for the Fe exports Sanders et al. (2015) argues that through their calculations small wetlands can provide ~ 1.8Tg / year equivalent to the riverine load of 1.5Tg/year.*

L213 - the authors may wish to look at Nick Ward's work on terrestrial-aquatic interfaces to better consider these points.

*Reply: The authors are familiar with some of Nick Ward's work. His work in the Amazon basin is quite interesting and a good comparison of large river systems compared to our much smaller systems. His excellent review "Where Carbon Flows" was a very useful reference for the introduction placing our work in the greater context. The study of carbon export across estuaries is in general understudied and we have added a sentence to this section (line 212 page 11) to highlight the gap he identified:*

"The study of carbon dynamics in estuaries in fact remains a key knowledge gap globally (Ward et al., 2017)."

Technical: L131: hurricane was misspelled; L222 - important in place of significant

*Reply: Addressed see the appropriate lines*

---

## Author Comment (AC2)

The manuscript, entitled "The response of small boreal catchments to extreme weather event: Hurricane Larry" by Heerah et al. evaluated DOM and iron concentration after 2021 Hurricane Larry in Newfoundland, and found that the forests and peatlands may have capacities to buffer DOC and colour, while wetlands may buffer the increase in iron concentrations. The study provides insights into impacts of extreme weather events on water quality in boreal regions. However, I do have some concerns and suggestions for improvement, outlined below.

**General comments:**

1. This study may have a major issue of pseudoduplication, which affects the validity of the statistical analyses within each watershed. For each watershed, only one time sample collection was conducted before and after the hurricane, respectively. Regardless of the number of samples collected per event, applying statistical analyses to compare the changes in colour, DOC, iron etc., before and after the hurricane within a single watershed is problematic because these samples are not independent. To address this, the study could combine all samples into two groups: samples from three watersheds before the hurricane vs. samples from three watersheds after the hurricane. This approach would mitigate the pseudoreplication issue and ensure more robust statistical results.

*Reply: We do not believe there is an issue of pseudoduplication in this paper. In our initial statistical analysis, we did combine the measurements into a before and after group and conducted t-test between them to assess statistical relevance, which was discussed on line 145.*

*Line 145 – "Both temperature and pH (Table 1) exhibited statistically significant decreases in all three catchments between sampling times (t-tests, p = 0.0017, p=0.0499, respectively)"*

*To address this miscommunication of statistical analysis we have moved line 145 to line 151. We acknowledge that without the new statistical analysis method section, the treatment of data was not clear. Based on reviewer 2 and reviewer 1 comments, the description for the statistical analysis has been found to not be clear. A new section, section 2.6(lines128-134, page 6) has been added to the methods to attempt to correct this miscommunication. This new section can be seen in the response to your method comments.*

*While compiling this information we re-examined our statistical approaches and have decided that utilizing a two tail - two sample t-test was not the most appropriate statistical test to assess the increase in parameters after the hurricane. Instead, we have now conducted a one tail- paired t-test and report these new results in this manuscript. These tests show statistical relevance in all parameters except DOC. The new results are discussed in section 3.1(lines 153-155)*

*Amended Lines 153-155: "Increases in DOC, iron concentrations and a350 were seen in all rivers as shown in Fig.3. Fe and a350 increases were found to be statistically significant (t-test, p =0.0132, p =0.0389, respectively) with DOC failing to achieve statistical relevance (p = 0.9821)."*

*Additionally, we do not believe there is an issue of pseudoduplication in this paper based on sampling time. While there are only two time points in this study in an environmental context the combination of time and water flow makes samples collected in the same watershed independent of each other. The water flowing through the river 24 hours after the last sampling time is considered independent of the previous sampling due to it being considered a different water mass. As the river*

*flows chemical properties can change as a result of weather changes and differences in stage height.*

2. Regardless of whether the above pseudoreplication issue is valid, another major concern is that some conclusions are not supported by statistically significant results. For instance, the study suggests that a high percentage of forest and peatlands buffered against increases in DOC and colour, with wetlands buffering an increase in iron concentrations. However, Table 2 does not indicate any statistically significant correlations supporting these statements. Similarly, the higher DOM properties observed in Figure 3 are not statistically validated.

*Reply: The lack of statistically relevant results has been pointed out in reviewer 1 comments. We agree with reviewer 1 that the small sample size greatly hinders achieving statistically relevant results and the discussion should rather focus on solely on the correlation strength. This is further emphasized by the statistically relevant increases in parameters discussed in the replies above. The discussion of table 2 has thus been amended to remove p values and focusing sorely on correlation strength. It is important to note that lines 158-160 gives the caveat that the results discussed did not meet the p value threshold. However, we strongly believe this is due to the low degrees of freedom used as opposed to there being no actual change in concentrations.*

**Specific comments:**

L47-50: Please briefly describe how forests and peat barrens regulate DOM and iron.

"Catchments in Newfoundland, Canada are dominated by peat barrens and boreal forest (Latifovic et al., 2017), both rich in organic carbon and iron. DOM from boreal and peatland ecosystems have been identified as important for biogeochemical nutrient cycling (Krachler et al., 2010; Kritzberg et al., 2014; Worrall et al., 2006). This DOM can form complexes with iron more resistant to flocculation allowing greater terrestrial inputs into the coastal environment (Heerah and Reader, 2022; Herzog et al., 2020; Krachler et al., 2015; Kritzberg et al., 2014)."

*Reply: DOM is formed from the breakdown of biological material in the environment. In the terrestrial environment the major source of biological material is the vegetation present on the land. In the boreal environment where peat and forest dominate as sources of vegetation, they will be the dominant inputs in the DOM pool. Peat and boreal forest thus serve to regulate DOM through its production of biomass. The regulation of iron from peat barrens and forest are slightly less direct. Terrestrial iron can come from either microbial sources or the chemical weathering of mineral in the soil and subsequent mobilisation into the waterways. The specific ways how boreal forest and peat barrens regulate the delivery of iron is a current knowledge gap in the literature as these environments are able to transport more iron into the marine environment than rivers from other regions. This is touched upon in line 50. Peat barrens regulate the concentration of iron as they are also anoxic environments, and Fe has been found to accumulate in anoxic environments as they can remain in the soluble Fe (II) form as opposed to precipitating out as an iron hydroxide. The soils in boreal forest have also been found to be acidic than temperate forest allowing more Fe (II) to exist in the soils. Lines 49 -54 have been added to the manuscript to address this comment.*

"*The acidic soils in boreal forest and the anoxic conditions present in peat barrens can allow for the accumulation iron in the soils (Abesser et al., 2006; Kaal et al., 2022). DOM from boreal and peatland ecosystems have been identified as important for biogeochemical nutrient cycling (Krachler et al., 2010;*

*Kritzberg et al., 2014; Worrall et al., 2006). Peat and boreal forest are the dominant sources of biological material to the DOM pool in these environments producing DOM enriched in aromatics, lignins, humic acids and humic ligands (Krachler et al., 2005; Worrall et al., 2006)."*

Section 2.1: How many samples were collected before and after the hurricane in each stream?

*Reply: This information has been added into section 2.1 at line 88.*

*"Two 500 ml acid washed polycarbonate bottles were collected at each site before and after landfall. The bottles were transported on ice back to the lab where they filtered the day of collection ashed GF/F filters (Whatman, nominal pore size 0.7 μm). The filtered sample was then subdivided for dFe, DOC, and $a_{350}$, measurements. BOD samples were conducted with un-filtered sample water."*

L84-85: Can you show the gauged sizes of those three watersheds?

*Reply: The description of the watershed sizes and their landcover breakdown is discussed in section 2.2 on lines 103-105. Seal Cove brook is 53.6 km², South River near Holyrood is 17.3 km², and St Shotts River near Trepassey is 15.5km².*

Figure 1: Can you insert three sub-figures to show land use types of each watershed?

*Reply: The three watersheds land cover is broken down in section 2.2 in lines 103-106 focusing on the relevant land cover types discussed in this study. This information can be added as a table if the reviewer finds it pertinent information to include. The authors believe adding this data visually to Figure 1 may reduce readability of the figure. Lines 103-106 do not give the full breakdown of the landcover but cover 86.3% of SC's catchment, 80.02% of SR's and 86.02% of SS's. The remaining breakdown is outlined below:*

| Landcover Type | Seal Cove | South River | St. Shott's |
|---|---|---|---|
| Lichen- moss | 0.40% | 3.31% | 1.68% |
| Barren land | 0.04% | 0.90% | 0% |
| Water | 12.87% | 7.36% | 11.77% |
| Urban | 0.33% | 2.32% | 0.52% |
| Grassland | 0% | 0.096% | 0% |

L104: Include the relevant equation. In addition, I did not see any analyses about this index.

*Reply: The relevant equation has been added below Line 115. See the following equation below as well.*

$$RB\ index = \frac{\sum_{i=1}^{n}|q_i - q_{i-1}|}{\sum_{i=1}^{n} q_i}$$

*The result of the indexes was added to Table 1. Rather than reproduce the table in this reply. The numbers are as follows South River (0.3872), Seal Cove (0.37722), and St. Shotts (1.5719). A short*

*discussion of the results of the index has been added and can be found in Line 159- 160 and lines 190-194.*

*Lines 159 -160: "Discharge is fairly low not exceeding 1m3/s at any point in the sample period (Table 1). The RB index can be used to compare the river's response to storm events."*

*Lines 190-194: "The difference in flashiness is expected based on the size, land cover differences and catchment complexity (Baker et al., 2004). SC, as the largest of the three catchments was the least flashy incorporating the massive increase in precipitation. SR had the highest increases in fluxes of material following landfall despite being the second largest catchment. The dramatic increase in flux highlights that in addition to catchment size, landcover plays a major role in catchment response."*

Methods section:

(1) I would suggest adding a subsection to provide more information about the Hurricane Larry, for example, wind speed, amount the rainfall, etc. This information helps understand the severity of the hurricane and its consequence on water quality.

*Reply: This section was added at line 67.*

*"Hurricane Larry reached Newfoundland as a category 1 hurricane with a maximum sustained wind speed of 120km/hr and gusts reaching a maximum of 180km/hr. Rainfall was around 25 to 35mm over a very short period of time causing localized flooding with significant storm surges with waves reaching 3.6 m (Brown, 2021)"*

(2) What statistical methods were used?

*A section describing the statistical tests used has been added section 2. 6, based on earlier comments going from line 142-148, page 7.*

*"2.6 Statistical Analysis*

*All statistics were carried out in MATLAB 2020A. One tail paired t-tests were conducted on each parameter for samples before and after landfall, catchments were combined forming a before and after group. T-tests were conducted using a confidence interval of 0.05. The t-test showed if the changes observed in parameters were statistically different from one another. The Δ of parameters were compared with land cover using Pearson's correlations test. The Δs for each parameter were grouped together along with an individual landcover type to carry out the analysis, reducing statistical power, the ρ for the correlation analysis is reported but p-values are not due to the small sample size and low degrees of freedom.*

L141: this should be Table 1.

*Reply: addressed*